# A Comparison of Multiple Odor Source Localization Algorithms

**DOI:** 10.3390/s23104799

**Published:** 2023-05-16

**Authors:** Marshall Staples, Chris Hugenholtz, Alex Serrano-Ramirez, Thomas E. Barchyn, Mozhou Gao

**Affiliations:** 1Centre for Smart Emissions Sensing Technologies, Department of Geography, University of Calgary, Calgary, AB T2N 1N4, Canada; 2Department of Mechanical Engineering, University of Calgary, Calgary, AB T2N 1N4, Canada

**Keywords:** remote gas sensing, gas source localization, mobile robot olfaction

## Abstract

There are two primary algorithms for autonomous multiple odor source localization (MOSL) in an environment with turbulent fluid flow: Independent Posteriors (IP) and Dempster–Shafer (DS) theory algorithms. Both of these algorithms use a form of occupancy grid mapping to map the probability that a given location is a source. They have potential applications to assist in locating emitting sources using mobile point sensors. However, the performance and limitations of these two algorithms is currently unknown, and a better understanding of their effectiveness under various conditions is required prior to application. To address this knowledge gap, we tested the response of both algorithms to different environmental and odor search parameters. The localization performance of the algorithms was measured using the earth mover’s distance. Results indicate that the IP algorithm outperformed the DS theory algorithm by minimizing source attribution in locations where there were no sources, while correctly identifying source locations. The DS theory algorithm also identified actual sources correctly but incorrectly attributed emissions to many locations where there were no sources. These results suggest that the IP algorithm offers a more appropriate approach for solving the MOSL problem in environments with turbulent fluid flow.

## 1. Introduction

Locating the source of chemical releases is an important challenge in the field of robotics. Traditionally, human searchers and animals with better olfaction capabilities have been used to perform localization tasks for search and rescue, chemical leaks, and finding concealed narcotics and explosive devices such as land mines. However, there are many challenges inherent in this approach. Training humans and animals requires a large amount of time and effort; they have limited endurance, there is a limit to the set of chemicals they can sense [1], and there are situations in which the safety of human and animal searchers precludes their application. As a result, the application of robots with advanced olfactory and navigation capabilities has emerged as a machine-based alternative. One of the major research challenges with robots in olfactory applications is the efficient and effective localization of multiple unknown sources in outdoor environments under turbulent conditions. This type of situation commonly arises with fugitive chemical emissions in industrial settings.

Chemical or odor dispersion occurs through two primary mechanisms: diffusion and advection. Diffusion occurs due to a difference in chemical concentration. When diffusion is the dominant dispersion mechanism, chemical gradients form, with concentration decreasing with distance from the source. Initial research took advantage of this fact and developed algorithms to follow the chemical gradient to its source. These algorithms are thus classified as ‘Chemotaxis’ algorithms. Examples include the E. Coli, sometimes called the Biased Random Walking (BRW) algorithm [2], and gradient-based algorithms [1,3].

In outdoor environments, advection, not diffusion, is the dominant form of dispersion [4]. Chemotaxis algorithms typically rely on smooth chemical gradients to find the odor source. Under turbulent airflow conditions, research has shown that Chemotaxis algorithms have a much lower success rate than Anemotaxis algorithms, which use fluid flow measurements along with chemical concentration measurements [1,5]. The odor plume under turbulent conditions typically exhibits spatio-temporal variability and does not follow a smooth chemical gradient. To overcome fluctuations in the concentration gradient, Anemotaxis algorithms such as Silkworm moth [3], Zigzag [6], and Dung Beetle use the wind flow direction to follow the plume upwind to the source.

The aforementioned algorithms are forms of reactive algorithms. One of the shortcomings of reactive algorithms is that under dilute conditions, detectable events are separated by gaps with no detectable anomaly. As the delay between detections increases, the performance of reactive algorithms decreases [7]. Cognitive algorithms differ from reactive algorithms in that they use information from current and past measurements to guide the searcher’s movements. Even when a plume is dilute and very few odor detections occur, cognitive algorithms are still able to find the odor source, as each odor detection is used to improve the belief state of the possible leak source location. Several different methods have been developed to retain knowledge of the current state, including concentration grid maps [8,9] and the ‘Source Likelihood Map’ [9].

The above algorithms only consider one odor source; however, multiple odor sources are common in many applications such as fugitive chemical emissions from industrial facilities. Typically, the plume is detected far downwind of the facility, and it is difficult to ascertain if the plume originates from a single source or multiple sources. In this case, multiple odor source localization (MOSL) algorithms need to be used. MOSL algorithms for mobile platforms currently fall into three categories: occupancy grid mapping algorithms [10,11,12,13], particle swarm optimization (PSO) algorithms [14,15,16,17], and learning algorithms [18,19]. In [19], a sampling strategy for a multi-robot system using sparse Bayesian learning was presented and tested. When testing was performed with multiple sources, the same model was used for both the simulation environment and the model-based exploration strategy. The sampling strategy was also tested using a more complex simulation environment for the case of a single source to determine the effects of model mismatch, and a drop in source localization was observed. The localization performance comparison used the earth mover’s distance (EMD) as the metric. In [19], reinforcement learning along with existing domain knowledge was used to develop a policy for the movement of a searcher for the problem of localizing multiple sources in a steady-state environment. Domain knowledge was developed using the methods from [18]. The Asynchronous Actor Critic (A3C) algorithm was used, various reward functions and observation spaces were tested, and the localization performance was compared using the EMD.

In [14], a modified particle swarm optimization (MPSO) technique for multiple sources was used that relied on turning a plume off after finding the source so that the next source could then be searched for. The ability to turn off plumes is a major assumption and is unrealistic in many real scenarios. A niching strategy was used in [15] to split and merge the particles into niches to ensure each niche of particles followed different plumes to the sources, which parallelizes the search even in situations with multiple peaks. A Diverse-PSO that uses group dynamics was presented in [16] and demonstrated better performance than MSPO and Niching-PSO in a turbulent flow environment with multiple sources. In [17], a hybrid teaching learning particle swarm optimization (HTLPSO) was presented, and its plume localization performance was compared to a PSO algorithm and a teaching–learning-based optimization (TLBO) algorithm.

Unlike PSO methods and the learning-based methods, occupancy grid mapping algorithms are not reliant on a specific path for the searcher(s) for localization, nor do they require multiple searchers like the PSO methods and [18] do. However, the relative performance of the IP and DS algorithms are not well understood, and particularly so in conditions with natural time-varying flow. Existing testing has been limited. For example, the IP algorithm has been tested in two situations in a weak flow environment, i.e., in a hallway and in an underwater environment. The DS algorithm has been tested in a single simulation setup and outdoor environment. Before these algorithms can be applied to localize fugitive emissions outdoors, further testing is needed. The IP algorithm has not yet been tested in time-varying environments [10,11], and the DS theory algorithm has been tested three times outdoors, but only the results from one of those experiments were presented [12,13].

To improve understanding of the performance of these algorithms in outdoor environments, we tested and compared the IP and DS algorithms under time-varying flows and a variety of conditions and evaluated their source-localization performance. We used a simulation approach with a filament dispersion model based on Farrell et al. [20]. This model was chosen due to its ability to simulate dispersion with time-varying wind flow and its fast computation speed. The localization performance of the two algorithms was measured using the earth mover’s distance (EMD). Using the EMD, we evaluated how localization performance responded to changes in environmental parameters, which included search area size, source position, mean wind speed, puff random walk, bandwidth, and release rate, as well as the searcher parameters—searcher velocity and sample frequency—which were also varied. In this work, we refer to searchers as those entities tasked with discovering the source of a chemical release or emission. Each search was performed using a raster search, and both algorithms used the same wind, concentration, and non-detection/detection concentration threshold. In each set of simulations, a single parameter was varied, with the parameter being either at its high or low level. This was conducted to determine the sensitivity of the algorithms to each parameter.

## 2. Algorithm Description

While both MOSL algorithms are similar in that they use occupancy grid (OG) mapping, there are two primary differences. First, the algorithms differ in how they represent the belief state. The IP algorithm uses probability theory as its belief representation [10,11] for each cell in the occupancy map, while the DS algorithm uses DS theory [21] instead of probability theory. Second, they differ in how the belief update is performed when there is a detection event. The DS algorithm assumes the conditional independence of detections, which means that each cell being updated is independent, so information from other cells is not considered during the probability update. By contrast, the IP algorithm assumes each cell is dependent for detection events, in which case, it considers information from other cells during the probability update.

When a detection occurs with multiple potential sources present, no clear information on possible upwind sources can be obtained. To address this, the IP algorithm uses an update rule for detection events where the cells are dependent. For the case where it is known that there is only a single source, a detection event provides information about the entire map [10,11]. In the case of multiple sources, a detection event can occur in response to one or more sources. The number of sources contributing to that detection event cannot be known without further information. Dependencies arise between the occupancy grid map cells associated with the possible source locations that could have contributed to the detection event. At each time step, the information from the measurement needs to be used to update the current occupancy grid map. To perform this grid map update, all dependencies need to be accounted for. The number of calculations needed to account for all the dependencies scales at $2^{C}$, where C is the number of cells in the occupancy grid map. This formulation quickly becomes intractable as the search area grows or the grid resolution increases.

Addressing the problem of exponential scaling with search area, the IP algorithm uses an update rule (Algorithm 1) that can handle multiple odor sources and scales linearly with number of grid cells. The IP algorithm calculates an approximation of the marginal posteriors of the cells within the occupancy grid map. The IP algorithm assumes the independence of the posterior probability and interprets the current measurement considering past measurements but does not reinterpret past data based on new information. Each update interprets the new marginal probabilities using the dependencies from the current measurement but does not maintain the dependencies after the measurements have been incorporated and become the new prior. In its updates, the IP algorithm does not reinterpret past data considering new data. While conducting this reduces computational complexity, this algorithm is path-dependent [11]; therefore, the resulting map depends on the order in which the measurements are taken, and early interpretations can disproportionately affect the result.
**Algorithm 1** IP algorithm ($\{l_{t-1,i}\},\left\{P_{i}^{t}\right\},\left\{z_{t}\right\}$) [11]1: Time starts at instant t=1
2: **For all** cell $m_{i}$ **do**3:   **if** $m_{i}$ is in the perceptual field of $z_{t}$ **then**4:     **if** $z_{t}=1$ ($z_{t}$ results in a detection)5:       ${\tilde{P}}_{i}^{t}=\frac{e^{l_{t-1,i}}}{1+e^{l_{t-1,i}}}$
6:       $l_{t,i}=\log\frac{1-\left(1-P_{f}^{t}\right)\left(1-P_{i}^{t}\right)\prod_{s\neq i}\left(1-P_{s}^{t}{\tilde{P}}_{s}^{t}\right)}{1-(1-P_{f}^{t})\prod_{s\neq i}\left(1-P_{s}^{t}{\tilde{P}}_{s}^{t}\right)}+l_{t-1,i}$
7:     **else** ($z_{t}=0$, $z_{t}$ results in a non-detection)8:       $l_{t,i}=\log\left(1-P_{i}^{t}\right)+l_{t-1,i}$
9:     **end if**10:   **else**11:     $l_{t,i}=l_{t-1,i}$
12:   **end if**13: **end for**14: return $\left\{l_{t,i}\right\}$
where $l_{0,i}=\log\frac{p\left(m_{i}\right)}{1-p\left(m_{i}\right)}$. It is trivial from $\left\{l_{t,i}\right\}$ to recover $p(m_{i}|z_{1:t})$. In fact $p\left(m_{i}|z_{1:t}\right)=\frac{e^{l_{t,i}}}{1+e^{l_{t,i}}}$


To address the problems associated with the search path dependence as described in [11], DS theory was used in [12] instead of probability theory. For the task of localization, each cell has two states: S (source) and $\bar{S}$ (not a source), which compose the frame of discernment θ. In probability theory, the entire belief is split between the elements making up the frame of discernment. DS theory, on the other hand, assigns belief to the power set of the frame of discernment ${\Theta =2}^{\theta }$. This power set for the given frame of discernment results in the following power set: $\left\{S,\bar{S},\left\{S,\bar{S}\right\},Ø\right\}$. The basic probability assignment (BPA), or $m\left(e\right)$, is the measure of available evidence for a given state in the subset $e\in 2^{\theta }$. The BPA assigned to $\left\{S,\bar{S}\right\}$ represents the amount of ignorance about the state of the cell. Following the convention of [21], the state $\left\{S,\bar{S}\right\}$ will be replaced with U for the unknown state. In its update rule, the DS algorithm ignores any potential dependencies between cells and assumes they are independent. The belief mass functions for detection and non-detection events are given as:(1)$$m\left(e\right){|}_{i}^{D}=\left\{\begin{array}{cc} \zeta p_{i} & e=S\\ 0 & e=\bar{S}\\ 1-\zeta p_{i} & e=U \end{array}\right.$$
(2)$$m\left(e\right){|}_{i}^{\bar{D}}=\left\{\begin{array}{cc} 0 & e=S\\ \zeta \left(1-\varepsilon \right)p_{i} & e=\bar{S}\\ 1-\zeta \left(1-\varepsilon \right)p_{i} & e=U \end{array}\right.$$
where m is the mass belief, e is the evidence of a given state, $p_{i}$ is the single source detection probability for cell *i*, ζ is the reliability of the gas transport model, D is a detection event, $\bar{D}$ is a non-detection event, S is the evidence that i is a source, $\bar{S}$ is the evidence that i is not a source, U is the evidence that i is the unknown state and is the set of both S and $\bar{S}$ ($U=\{S,\bar{S}\}$), and ϵ is the false-negative probability of a non-detection event. The reliability of the gas transport model ζ is a term that is unique to [12,13]. A conservative estimate for ζ is used in [12], whereas in [13], the value of ζ is thought to be a function of distance and the relationship was empirically derived. In this work, ζ was selected as 0.8 and the odor patchiness was chosen as 0.75. The OG map for the DS theory algorithm updates each cell independently. The three possible states S, $\bar{S}$, and U result in three different mass beliefs for each cell in the occupancy grid map. The mass belief update is given as:(3)$$m_{1,2}=\left(m_{1}⨁m_{2}\right)\left(e\right)=\sum_{e_{1}\cap e_{2}=e\neq Ø}\frac{m_{1}\left(e_{1}\right)m_{2}\left(e_{2}\right)}{1-K}$$
where K represents the amount of conflict between the two pieces of evidence and is given by:(4)$$K=\sum_{e_{1}\cap e_{2}=Ø}m_{1}\left(e_{1}\right)m_{2}\left(e_{2}\right)$$

The DS theory algorithm in [12,13] uses the air mass path (AMP) [22] to determine the range of possible locations where the air passing over a searcher could have come from. The AMP was also used to calculate the probability of the various paths that could have been taken by the air mass to reach the searcher’s position (Figure 1). The path is created using the history of wind velocities. The AMP probability calculation is restricted to only the locations within the path to reduce computation time. The edges of the path are found using a threshold-detection probability. The width of the path is a function of the variability in wind velocity and time expired prior to the present measurement. The probabilities obtained by the AMP are then used as an input to the algorithms as the single-source-detection probability. The full AMP formulation used here is described in [12,13]. The IP algorithm in [11] was tested in an environment with no strong airflow, and the update rule used did not take advection into account. For situations with strong airflow, the IP algorithm uses the AMP as its single-source-probability model. 

## 3. Materials and Methods

The experiments performed were set up using sensitivity analysis principles. The goal was to evaluate how the localization performance of the algorithms responded to changes in environmental/simulation and searcher parameters. The set of environmental and searcher parameters that varied, together with their various levels, is shown in Table 1. Each of the eight variables was tested at a high and low parameter value (level). Each level was tested 32 times for a total of 512 simulation runs. When a variable was not being tested at a high or low level, its value was set to its normal level (Table 1). An additional set of 32 simulations was performed at the normal parameter level for all parameters to be used as a reference set. The level of each parameter was selected to test the limits of the performance of the algorithms. For many of the parameters, such as mean wind speed, bandwidth, random walk, and release rate, we first conducted preliminary tests to determine the functional limits of the algorithms and simulator; testing at levels beyond those chosen would cause either both algorithms or the simulator to consistently fail.

Algorithm testing was performed using the filament-based dispersion model from [20]. This model is a computationally efficient model that generates a temporally and spatially varying wind field, which is used to advect the odor filaments. The simulator generates plumes that wag back and forth with changes in the wind field and produces relative dispersion along the instantaneous plume center line. This simulator is based on the same simulation used in [12]. The simulation in [20] was originally written for a single source and was modified to accommodate multiple sources in this work.

### 3.1. Experimental Setup

The simulation was divided into two main parts: the environment and the searcher (Figure 2). While the environment simulation was based on the work in [20], many modifications to the simulator were made, including extensive modifications to the dispersion model so that multiple odor sources could be used. The simulation begins by simulating the time-varying wind field using randomized boundary conditions as an input. The dispersion model takes the wind field $U_{x}U_{y}$, source locations $L_{sources}$, and release rates $R_{sources}$ as inputs. The dispersion model generates puffs of odor and models their dispersion. Next, the odor concentration of the puffs is calculated based on the searcher’s position $L_{robot}$ and the positions of each puff {$\vec{P}.x$, $\vec{P}.y$, $\vec{P}.z$}. The concentration at the searcher’s position is represented by Con(x,y) in Figure 2. The searcher’s position follows a raster search pattern and is updated by the searcher simulation. The times that measurements are taken and recorded are based on the measurement frequency. The wind velocity, concentration, simulation time, and searcher’s location are all recorded at each measurement step. The wind velocity is measured with sensor noise modeled by using additive Gaussian noise with zero mean and a standard deviation of 0.1 m/s. These variables are then used to calculate the AMP $P(L_{*}|L_{robot})$ and whether a detection or non-detection event occurred, represented by the binary variable $c_{x,y}$. This information is used by the MOSL algorithms to update their respective OG map. Both the dispersion and wind simulations were run with a 10 ms integration time step. All programming was conducted in Python 3.8.

#### 3.1.1. Wind Model

The wind simulation was based on the dynamic wind model in [20], which is given as:(5)$$\frac{\partial \bar{u}}{\partial t}=-\bar{u}\frac{\partial \bar{u}}{\partial x}-\bar{v}\frac{\partial \bar{u}}{\partial y}+\frac{1}{2}K_{x}\frac{\partial^{2}\bar{u}}{\partial x^{2}}+\frac{1}{2}K_{x}\frac{\partial^{2}\bar{u}}{\partial y^{2}}$$
(6)$$\frac{\partial \bar{v}}{\partial t}=-\bar{u}\frac{\partial \bar{v}}{\partial x}-\bar{v}\frac{\partial \bar{v}}{\partial y}+\frac{1}{2}K_{x}\frac{\partial^{2}\bar{v}}{\partial x^{2}}+\frac{1}{2}K_{y}\frac{\partial^{2}\bar{v}}{\partial y^{2}}$$
where $\bar{u}$ is the wind velocity in the *x* direction, $\bar{v}$ is the wind velocity in the y direction, and $K_{x}$ and $K_{y}$ represent diffusivity. The wind field was calculated at each time step using the 2D Lax–Wendroff method to solve Equations (5) and (6) numerically. The wind model’s inputs were the wind field at the previous time step and the boundary conditions at the current time. The wind velocity at the four corners of the wind field was generated at every time step using colored noise and the mean wind velocity. The colored noise inputs at the four corners generated the time- and spatial-varying wind field. The colored noise was generated by taking white noise and filtering it with a second-order filter. The bandwidth of the filter was chosen to be one of the simulation parameters to be tested because it influences the wind field variability. The wind velocities for the boundary of the wind field were calculated by linearly interpolating between each of the corner velocities. The nodes within the wind field were spaced at 10 m intervals. This scale was used to simulate turbulent eddies that were larger than the width of the plume.

#### 3.1.2. Dispersion Model

The dispersion simulation used a filament-based dispersion model. Within the model, odor is represented by filaments, each of which has a position and radius. The dispersion of the filaments occurs through three main mechanisms: advection due to the wind field, filament random walk, and increasing filament radius, as shown in Figure 3. Advection is the dominant mode of dispersion and simulates dispersion due to large-scale wind eddies that cause plumes to meander. The wind velocity at the position of each filament is calculated by interpolating between the velocity of the surrounding nodes in the wind field. The new position of the filament is then calculated by integrating the wind velocity and adding a random walk. The random walk is a Gaussian-distributed random variable with zero mean and a non-zero standard deviation. The standard deviation of the random walk was another simulation parameter that varied. The model uses random walk to simulate the dispersion due to medium-scale turbulent eddies [20], where the scale of the random walk is similar to the width of the plume. The random walk controls the relative dispersion of the filaments about the plume centerline. The third mechanism of dispersion is the growth of the radius of each filament and is used to model dispersion due to molecular diffusion and the smallest length scales of the turbulent flow. The total mass of each filament is constant, but the density of the odor within the filament decreases as its radius increases. The initial radius of the puffs was 1.1 m, and the growth rate was 0.01 m/s. Concentration is calculated by summing the total mass of filaments within the sensing volume of the searcher.

#### 3.1.3. Source Configuration

Three different source configurations were used: (i) crosswind, (ii) staggered, and (iii) inline, with the mean wind direction shown in Figure 4. Each source configuration results in the plumes from the two sources interacting uniquely. For the inline source configuration, when there is minimal plume meander, the plumes fully overlap. In the staggered configuration, the two plumes partially overlap. In the crosswind configuration, the two plumes are fully separated. The assumption of the configurations creating overlapping, partially overlapping, and fully separated plumes is only valid for cases where the mean wind velocity is relatively constant and plume meander is minimal.

#### 3.1.4. Searcher Simulation Description

In each simulation, the searcher started the simulation at position (0,0). The searcher then performed a raster search by moving along the y-direction towards the far side of the search area. Once at the far side, the searcher then moved upwind and traversed the search area back to the side where it started. This pattern was repeated until the searcher reached the upwind side of the search area and ended the simulation. The search area was 40 m by 40 m, and sources did not exist outside the defined search area.

The search model is designed such that the searcher does not directly measure detections and non-detections. Instead, the searcher measures concentration, which is translated into either a detection or non-detection event. In [10,11], a constant concentration threshold was used to determine detections and non-detections. A variable threshold concentration was used in [12,13] to determine detections vs. non-detections. In this work, a static concentration threshold was used to identify detections for simplicity.

Two separate emissions sources were used in each of the simulations performed for this work. There were three different configurations for these two sources. The first was the inline configuration, where the two sources were aligned with the mean wind direction, with one source 15 m upwind of the other. The second was the staggered configuration, with one source 10 m downwind and 5 m crosswind from the other. The final was the crosswind configuration, where the sources were in the same position relative to the mean wind direction and separated by 8 m crosswind. We postulated that the inline source configuration would be the most difficult localization task, as the downwind source would be within the plume of the upwind source.

One of the simulation parameter levels varied was the searcher position in the mean wind direction. The reason for this selection was to test how the number of plume intersections influences localization. At the normal parameter level, the sources were centered in the middle of the search area. At the low source position parameter level, the sources were moved downwind from their position at the normal parameter level. When the sources are further downwind, there are fewer plume crossings, as the plume has less distance to travel before it exits the search area. Conversely, the high source position parameter level, where the sources are upwind from their position at the normal parameter level, can result in an increased number of plume crossings. The positioning of the plumes can also affect the maximum crosswind distance that a plume meanders, as there is more time from when a puff is released until it exits the search area. This is also dependent on the wind fields that advect the plume.

### 3.2. Evaluation Metric

To evaluate and compare the results for each MOSL algorithm, we used the EMD [19] as the localization performance metric. In the EMD, the marginal probability for each cell in each map can be thought of as mounds of dirt, and the EMD is the cost to move all the mounds of dirt in one distribution to make it the same as the other distribution. The cost is equal to the distance moved multiplied by the height of the distribution. The lower the EMD score, the more similar the two distributions are, with a perfect score being zero.

In our work, the total mass in the true source map and the estimated source map (the probability OG maps from the algorithms) are both normalized to a total mass of 1 so that both distributions have equal mass. The cost matrix is produced using the Euclidean distance matrix, which maps the distance from any point in one distribution to any point in the other. The DS theory algorithm has belief over three terms for each potential location $\left\{S,\bar{S},U\right\}$ in its OG map, as it is based on DS theory, and the IP algorithm has a single probability for each location in the OG map. In this form, the DS theory OG map cannot be directly compared to the IP algorithm’s OG map. To perform a comparison, the DS theory algorithm’s belief OG map needs to be converted to a probability OG map, which was conducted using the pignistic transform, which is the most common transform from DS theory to probability theory:(7)$$P_{Bet}\left[x\right]=\sum_{x\in e_{1}\subseteq \Theta }\frac{m\left(e_{1}\right)}{cardinality\left(e_{1}\right)}\forall x\in \Theta$$
where x∈Θ is a singleton in the power set Θ (either S or $\bar{S})$, $cardinality\left(e_{1}\right)$ is the number of elements in the set $e_{1}=\{S,\bar{S},U\}$, and $P_{Bet}\left[x\right]$ is the probability of proposition x.

## 4. Results

The performance of the two MOSL algorithms was tested in 1302 simulations. Three source-location configurations were used, where the plumes were (i) fully separated, (ii) fully overlapping, and (iii) partly overlapping. The algorithms were tested under a range of conditions by varying seven simulation and searcher parameters of interest (Table 1) and testing them at high and low levels. The various source configurations and parameter levels contextualize how the search algorithms perform in localizing unknown sources across a wide set of conditions.

### 4.1. Occupancy Grid Maps

In this section, the OG maps for the IP and DS theory algorithms are presented in Figure 5 and Figure 6. To establish a baseline, the OG maps for both algorithms at the normal parameter level and the three source configurations are presented in Figure 5.

The OG maps show the spatial distributions of the source probability estimates in Figure 5, which appear as bands of higher probability scores extending downwind from the sources. The spatial bands are narrower for the IP algorithm in all simulations. The bands show more spatial variability in the probability scores with the IP algorithm, which manifests as patchiness in each band.

For the DS algorithm, these spatial bands also extend upwind of the sources with wider lateral spread and lower source density compared to the spatial bands on the downwind side. The upwind sources with the DS algorithm were a result of how the mass beliefs were combined. With the DS algorithm, once belief has been assigned to a possible source location, the degree of belief in that location will no longer decrease even with subsequent non-detections. The upwind sources show possible source locations identified by the AMP during detection events. The lack of subsequent detection events resulted in the lower probability associated with each location when compared to the sources on the downwind side of the true source locations. On the downwind side of the sources, the possible locations had multiple detection events reinforcing the high probability score.

Two bands are apparent from each source in the crosswind source configuration with both algorithms, but merge into one band for the inline and staggered simulations configurations. The spatial band in the staggered configuration narrowed in its width between the two sources. With the DS algorithm, there were two spatial bands: one band with higher probability scores and source density downwind of the sources, and the other spatial band with lower density and lower probability scores extending upwind of the sources. Between the two sources, both the upwind band and downwind band narrow. The downwind band of sources extended from the source furthest upwind, and the upwind band extended from the source that was furthest downwind.

With the low release rate, the predicted sources often failed to predict the true source positions, as seen in Figure 6. The probability scores were far lower than with any other parameter level, and in many simulations, the IP algorithm’s OG map did not have any predicted sources. Most of the probability scores for the DS algorithm were similar to the other parameter levels; however, in some simulations, the probability scores were very low, which was similar to the IP algorithm. The sources with higher probability scores did not ever include the true source positions. The locations with lower probability scores occasionally included the true source positions.

The widest crosswind spread was observed with the low wind speed parameter level Figure 6. With the low wind speed parameter level, the two distinct bands that were typical of the crosswind configuration merged into a single band with abrupt crosswind shifts which reached the edges of the search area. At the low wind speed parameter level, the density of predicted sources and the probability scores decreased with the IP algorithm and increased with the DS algorithm.

At the high wind speed parameter level, the spatial bands were shorter and concentrated closer to the true source locations. The bands did not extend to the edge of the search area in either the downwind or upwind direction. The probability scores decreased the further they were from the source. The spatial bands were narrower in the crosswind direction when compared to other parameter levels. Unlike the other parameter levels with high wind speed, the IP algorithm had sparse spatial bands extending upwind of the sources.

### 4.2. Statistical Pairwise Tests

The median EDM scores for the given parameter levels and algorithms were compared in order to analyze how changes in parameter levels affect localization performance and the relative performance between the algorithms. The statistical significance for each pairwise test between the algorithms was calculated using the Wilcoxon signed-rank test. The Wilcoxon signed-rank test was selected because it is used for two dependent samples and it is a non-parametric test, meaning that no assumptions would be made about the distribution of EMD scores. The Mann–Whitney U test was used to determine the statistical significance of the difference between the distributions of EMD scores between the high and low parameter levels for a given algorithm. The Mann–Whitney U test is typically used to compare the difference between two independent samples, and, like the Wilcoxon signed-rank test, it is non-parametric. For each source configuration, the median EMD scores for each parameter level and algorithm are shown in Table 2 and Table 3, along with the differences in median values between the high and low level for each algorithm, the difference in median scores between the algorithms at each level, and the associated *p*-value for each comparison. Statistical significance was evaluated with a *p*-value < 0.01 for all tests.

For most parameter levels, the IP algorithm typically outperformed the DS algorithm in terms of having a lower EMD score, as shown in Table 2. The DS algorithm only outperformed the IP algorithm at the low release rate across all parameter levels. There was not any statistical difference between the two algorithms at the low source position in the staggered configuration. The largest difference in EMD score between the algorithms was at the low sample frequency parameter level and the high wind speed parameter level for all configurations.

The largest change in median EMD scores between the low and high parameter level was with release rate, and this result consistently occurred with all configurations and both algorithms, as shown in Table 3. The largest difference in median EMD score between the high and low parameter level occurred with the IP algorithm with the inline configuration between the high and low release rate. For most parameters, the low level typically had a greater EMD score than the high level. The parameters where the high level had the higher EMD score were wind speed and random walk for only the DS algorithm for all source configurations, and source position for both algorithms with the crosswind source configuration. There was not any statistical difference between the high and low parameter levels for searcher velocity with both algorithms and all configurations, random walk with the IP algorithm and all configurations, or sample frequency with the DS algorithm and the inline configuration.

### 4.3. Violin Plots of the EMD Score

The EMD scores for each algorithm and parameter level are presented in Figure 7, with each source configuration represented by a separate violin plot. The width of a violin plot is related to the frequency of an EMD score for a given parameter level. The range in the vertical direction represents the range of observed scores. The black bars on the violin plots represent the range of scores that lie between the 25th percentile and 75th percentile for the given parameter level. The white dots represent the median EMD value.

Generally, the EMD score results were similar between the three source configurations, as shown in Figure 7. The median EMD scores for both algorithms in the staggered source configuration were generally higher than the median EMD scores for the other two source configurations. The median EMD scores were higher for the DS algorithm than the IP algorithm, indicating that the IP algorithm typically performs source localization better than the DS algorithm in terms of the median EMD score. The only exception was the low release rate, where median EMD scores were consistently higher for the IP algorithm. Median EMD scores of both algorithms for the low release rate parameter level were higher than the other parameter levels, except for the high source position parameter level in the crosswind configuration.

Within each source configuration, the DS algorithm’s median EMD scores for each parameter level were relatively similar to each other, whereas the IP algorithm’s median EMD scores had higher variance between parameter levels. The IP algorithm also had higher variance in the EMD scores between simulations with the same set of parameter levels, which was observed because of the greater difference in scores between the 25th and 75th percentile for the IP algorithm and the longer the tails of the violin plots compared to the DS algorithm. The higher variance in EMD scores indicates that the different spatial probability bands for the IP algorithm result in a change in EMD score, and the minimal variance in the spatial probability bands for the DS algorithm result in minimal changes in EMD score. In other words, changes in spatial distribution of probable sources result in significant changes in EMD scores.

The lowest EMD score was observed with high sample frequency in the inline configuration using the IP algorithm. The parameter levels where the lowest median EMD scores were obtained across all source configurations were high sample frequency and high mean wind speed (both with the IP algorithm).

## 5. Discussion

Overall, the IP algorithm outperformed the DS theory algorithm in terms of source localization. The IP algorithm’s predicted sources were closer to the true source locations than the DS algorithm’s for all parameter levels, with a few exceptions. The OG maps had bands of predicted sources that extended from the true source location to the downwind edge of the search area. The width of the bands of predicted sources were narrower for the IP algorithm than the DS algorithm. The density of the sources in the bands were greater when they were closer to the true source locations. Both algorithms overpredict the total number of sources, with the DS algorithm overpredicting the number of sources to a greater extent than the IP algorithm. All the predicted sources of the IP algorithm were subsets of the predicted sources of the DS algorithm. Additionally, the DS algorithm had predicted sources that were observed on the upwind side of the true source locations in the DS algorithm’s OG maps and were not observed with the OG maps produced by the IP algorithm. The IP algorithm also outperformed the DS algorithm in terms of EMD scoring, with a few exceptions, demonstrating consistency between the qualitative OG map observations and the quantitative EMD scores. The exceptions include the low source position parameter level in the staggered configuration, where there was not a statistical difference between the EMD scores, and the low release rate parameter level, where the DS algorithm had a lower EMD score than the IP algorithm. With the low release rates, the predicted sources often failed to include the true source location at low release rates with both algorithms. While the IP algorithm overall had lower median EMD scores, the variance in EMD scores was much higher than the DS algorithm. The difference between the 75th percentile and the 25th percentile EMD scores was greater for the IP algorithm than the DS algorithm. The range of observed scores for each parameter level was also greater. The IP algorithm was more sensitive to changes in parameter level than the DS algorithm and had greater changes in median EMD score based on changes in parameter level. The IP algorithm had the lowest median EMD score, which was with the high sample frequency with inline configuration. The parameter levels across all configurations with the lowest EMD scores were high mean wind speed and high sample frequency with the IP algorithm. The median EMD scores for the DS algorithm were relatively similar for most parameter levels, with the exception of the low release rate and low and high source positions parameter levels. Generally, the results were the same across all configurations. However, the median EMD scores were higher for both algorithms in the staggered source configuration compared to the other configurations.

The IP algorithm typically had a lower median EMD score compared to the DS algorithm because the IP algorithm’s OG maps had a lower number of predicted sources, and the predicted sources were typically closer to the true source locations. The primary reason for the lower number of predicted sources was that the DS algorithm assumed that all detection events were independent, whereas the IP algorithm assumed detection events were dependent. Both algorithms assumed that non-detection events were independent. When the DS algorithm updated cells independently during a detection event, the algorithm did not consider that the other potential source locations calculated using the AMP could also have been the source, resulting in the increased number of predicted sources. To minimize the overidentification of sources during detection events, the IP algorithm conditioned its current probability update received from the AMP on the previous OG map, implicitly considering previous measurements in its update. Interpreting current measurements in the context of past measurements which were implicitly held in the OG map resulted in the updates for detection events being context-dependent, which resulted in the IP algorithm being path-dependent [11]. Path dependence means that the order in which the measurements are received will have an impact on the final OG map. Note that past measurements were also not reinterpreted in the context of the new measurement.

Contextualizing detection events on the current map led to one of the primary drawbacks of the IP algorithm: early measurements influence the interpretation of later measurements. Early in the search, high-variance events, such as large amplitude plume meander or higher-variance random walk, have a greater influence on the IP algorithm’s EMD score compared to the DS algorithm’s EMD score. This was because high-variance events that occurred early in the search developed an OG map with predicted source locations with a skewed position, and all future measurements were conditioned on the skewed OG map. Furthermore, the IP algorithm had fewer total sources in its OG maps, which resulted in greater weighting to movement of potential source positions on the EMD score. The effect of having a lower number of predicted sources was highlighted by both algorithms at the low release rate parameter level, where the OG maps were very sparse and the greatest range in EMD scores were observed.

Potential source locations erroneously identified early in the search often do not end up in the IP algorithm’s final OG map. This was because subsequent non-detection events lowered the probability of the potential source locations. The IP algorithm required fewer non-detections than the DS algorithm to lower the probability of potential source locations. This was evident from the number of sources upwind of the true source locations in the DS algorithm’s OG maps and the lack thereof with the IP. Once the searcher was upwind of the true source locations, only non-detection events occurred. Each non-detection lowered the probability of any potential source locations upwind of the searcher that were identified as potential sources earlier in the search. The number of non-detections upwind of the true source locations was sufficient for the IP algorithm to eliminate potential sources upwind of the true source locations, but not for the DS algorithm.

Comparing the DS algorithm’s performance to the literature is difficult because of the limited number of experiments presented and missing environmental parameters for those experiments, making results difficult to replicate. The pattern of taking many non-detection events to lower the belief of potential sources was observed in the original papers [12,13], which detail the DS algorithm. Bands of cells identified as occupied by a source S were observed extending upwind from the true source locations early in the search in both [12,13]. Both of these papers also used the raster search pattern, but unlike our work, [12] completed multiple ‘rounds’, repeating the raster search pattern multiple times per search. In [12], the bands begin to shrink in size after five rounds of searching the entire search area. In [13], after the second round of searching the search area, the bands had grown in thickness and formed a large patch around each of the source locations. Results on further rounds were not shown. The results after multiple rounds of searches showed that the DS algorithm was capable of lowering the probability of upwind sources but required more non-detection events than are measured in several rounds of searching. This could be a result of using the Dempster rule of combination as the combination rule, which lacks robustness when handling conflicting evidence [23]. The use of a different combination rule that handles conflicting evidence better may result in better performance with the DS algorithm.

The IP and DS theory algorithms were most sensitive to parameter-level changes in release rate. This sensitivity was asymmetric, with the largest observed difference in median EMD score occurring between the low release rate and normal parameter level, and minimal difference in median EMD score between the high release and the normal parameter level. The asymmetry is a result of using binary detection events for source localization, as the searcher only cares whether the concentration is above the threshold concentration for detection. At the normal release rate and high release rate, the plume is continuous for both cases and the searcher would observe a similar number of detection events. The asymmetry arises when the release rate is low enough that a discontinuous plume forms with a concentration below the detection limit dispersed along the instantaneous plume centerline. Above this release rate, all release rates appear the same to the searcher, and below this release rate, non-detection events occur even when the searcher is directly where the instantaneous plume would normally be at a higher release rate, resulting in a sparser OG map for both algorithms. With the low release rate parameter level, the release rate was very low, and very few detections were observed by the searcher, which resulted in a very sparse OG map, which contributed to the greater variance in EMD score.

The IP algorithm had poorer localization performance compared to the DS algorithm at the low parameter level in terms of EMD score and qualitative observations of the OG maps. The IP algorithm’s potential source locations frequently failed to include the true source locations, while the DS algorithm’s potential sources occasionally did include the true source locations. At the low release rate, the effects of the IP algorithm’s path dependence were magnified, as the locations in which detection events occurred were very variable.

The only parameter where the median EMD score was not statistically significant was the low source position parameter level in the staggered configuration. However, the EMD score did not properly evaluate the localization ability of the algorithms at the low source position parameter level based on qualitative observations of the OG maps. From the OG maps, we found that the DS algorithm’s localization was worse than the IP algorithm’s. The DS algorithm had more predicted sources than the IP algorithm, with sources extending upwind of the true sources to the upwind edge of the search area, whereas all the IP algorithm’s predicted sources were concentrated between the true source locations and the downwind edge of the search area. Because of the upwind predicted sources, the maximum distance between a predicted source and the true source locations was far greater for the DS algorithm than the IP algorithm. As the DS algorithm’s downwind sources were very dense and, at the low source position, the downwind predicted sources were all close to the true source location, the lower number of upwind sources had a very small contribution to the overall EMD score, resulting in the DS algorithm having a non-statistically significant difference in median EMD score from the IP algorithm. This was a result of the EMD score not being able to directly compare OG maps with different mass and requiring the mass of the predicted sources to be normalized to equal that of the true source locations.

Further evidence of issues with the EMD score to evaluate localization performance at the low source position was demonstrated by the IP algorithm’s median EMD score in the low source position, for the staggered configuration was higher than its median EMD score at the normal parameter level, despite the fact that the predicted sources were fewer in number and closer to the true source location compared to the normal parameter level. At the low source position, the true source locations were close to the downwind edge, which resulted in the fewest number of sources of any source position, and for the IP algorithm, all of the predicted sources were near the true source location. At the normal parameter level, there were far more predicted sources and a larger distance between many of the predicted sources and the true sources than the maximum distance observed at the low source position parameter level. Further issues with using the EMD score were highlighted, with the IP algorithm’s OG maps at the high source position having a lower score than the low source position for the inline and staggered source configurations, despite having more sources, and with many sources being the furthest distance from the true source location than with any other parameter level.

The underlying mechanisms responsible for the IP algorithm’s sensitivity to changes in parameter level were the same as the mechanisms that caused the IP algorithm to have greater variance in EMD score compared to the DS algorithm, namely, the path dependence of the IP algorithm and the lower number of predicted sources. The sensitivity to changes in parameter level due to path dependence was demonstrated through the best localization performance occurring with the high sample frequency and high wind speed parameter levels. Both high wind speed and high sample frequency parameter levels minimized high-variance measurements. Higher sample frequency provides better resolution on the exact positioning of the plume. In the case of the IP algorithm, due to better information, early measurements were less likely to produce a skewed map and cause future results to be interpreted poorly. By contrast, the DS theory algorithm’s median EMD score did not improve to the same extent as the IP algorithms with higher frequency.

The high wind speed parameter level also produced low median EMD scores. This was due to the fact that higher wind speed produces narrower plumes with less variable concentration at the edges of the plume when compared to a lower wind speed, as there is less time for dispersion to occur for the same distance downwind. Due to the lower amount of meander and lateral dispersion at the high wind speed parameter level, the IP algorithm had fewer high-variance measurements and produced narrower bands of probability, resulting in a better EMD score compared to the normal or low wind speed parameter level.

In general, the difference in the DS theory algorithm’s median EMD score between the high and low parameter levels was less than the IP algorithms due to the consistently high occupancy in the DS algorithms OG map. With the high number of predicted sources, there was little variation in predicted sources from one simulation to the next. Furthermore, the high number of predicted sources reduced the relative weighting on the small changes in source position that did occur, resulting in minimal difference in EMD score between simulations.

With the staggered configuration, the total width of the band of probability is thicker than the other two source configurations, resulting in higher median EMD scores for all parameter levels. The staggered configuration was designed to produce partially overlapping plumes. The partially overlapping plumes produced the largest crosswind width of detection events, resulting in the greater thickness of the bands of probable source locations.

## 6. Conclusions

The IP and DS algorithm were both tested under a variety of simulation parameter levels and source configurations, with each simulation setup being tested multiple times. The plumes were turbulently advected and varied spatiotemporally. The IP algorithm outperformed the DS algorithm in terms of median EMD score for most parameter levels, except for low source position for the staggered source configurations where the difference was not statistically significant and the low release rate for all configurations. For the low release rate, the DS algorithm had the lower EMD score. The highest median EMD score was observed with a low release rate with the IP algorithm in the staggered source configuration. The best localization performance was with the IP algorithm with the high sample frequency with the inline configuration. Both algorithms consistently failed to only localize the true leak sources and drastically overpredicted the number of leaks, with the IP algorithm overestimating the number to a lesser extent. Both algorithms failed to identify the true source locations at very low release rates, with the IP algorithm failing more frequently than the DS algorithm. While far from perfect, the IP algorithm should be selected as the MOSL algorithm of choice, and great care should be taken when searching for plumes with very low release rates or for plumes with concentrations near the minimum detection limit of the concentration sensor.

## Figures and Tables

**Figure 1 sensors-23-04799-f001:**
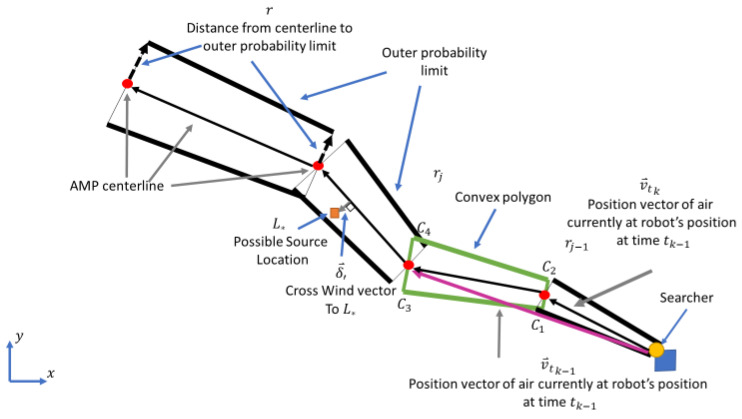
The airmass path.

**Figure 2 sensors-23-04799-f002:**
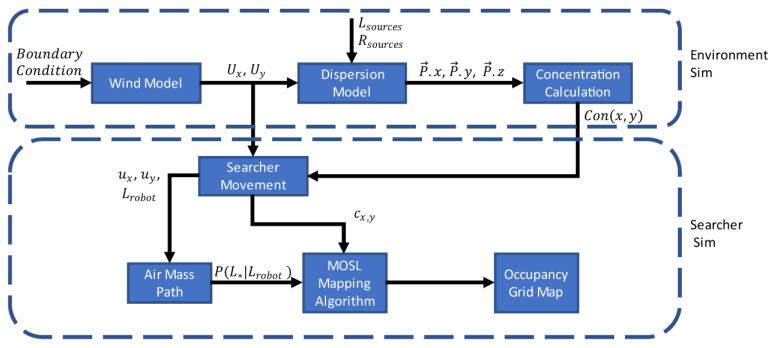
Configuration of the MOSL simulation.

**Figure 3 sensors-23-04799-f003:**
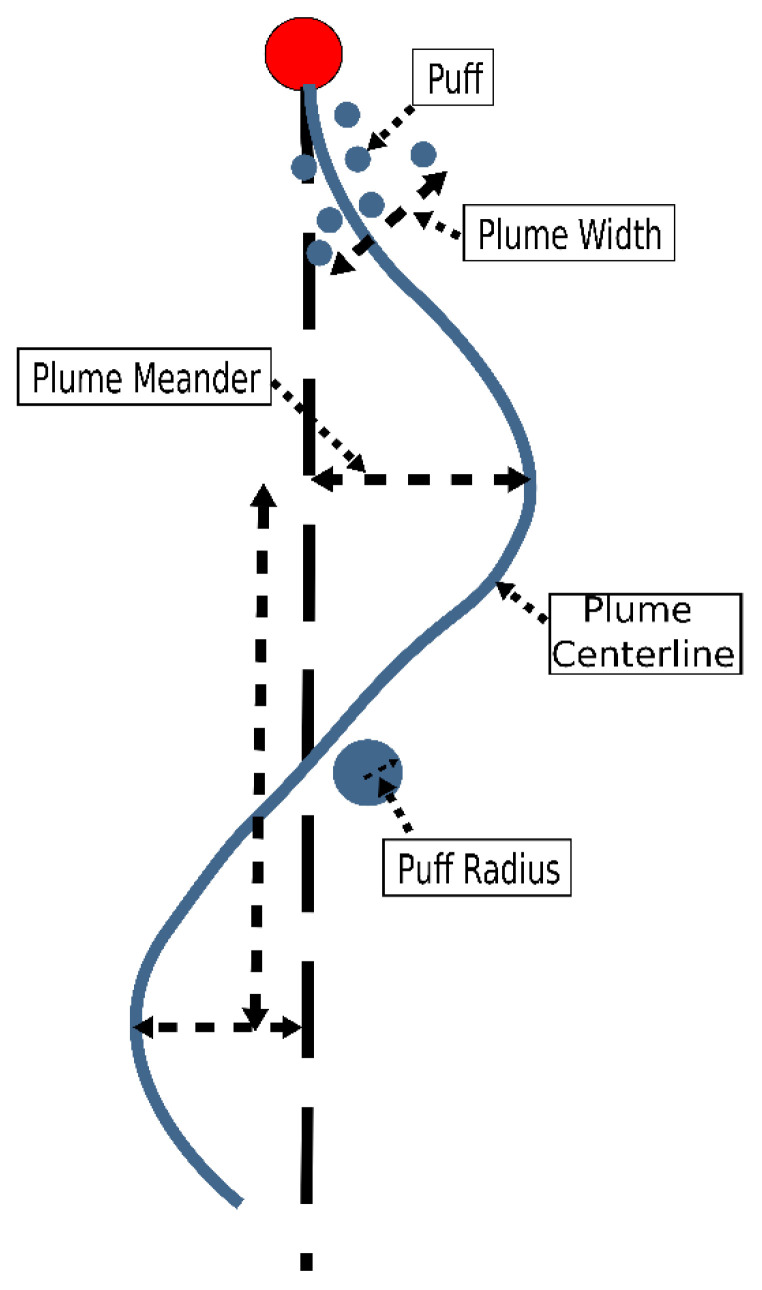
Plume dispersion model.

**Figure 4 sensors-23-04799-f004:**
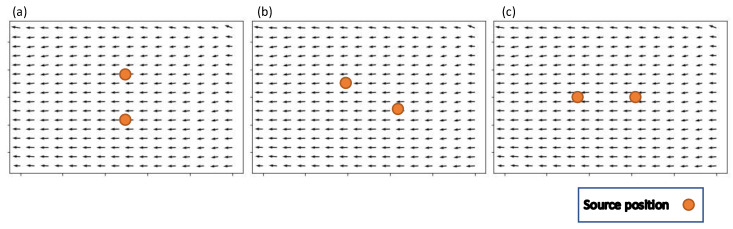
The source configurations with the wind field. The orange points indicate the source locations and the arrows represent the wind velocity with the length of the arrow representing magnitude. (**a**) crosswind source configuration, (**b**) staggered source configuration, (**c**) inline source configuration.

**Figure 5 sensors-23-04799-f005:**
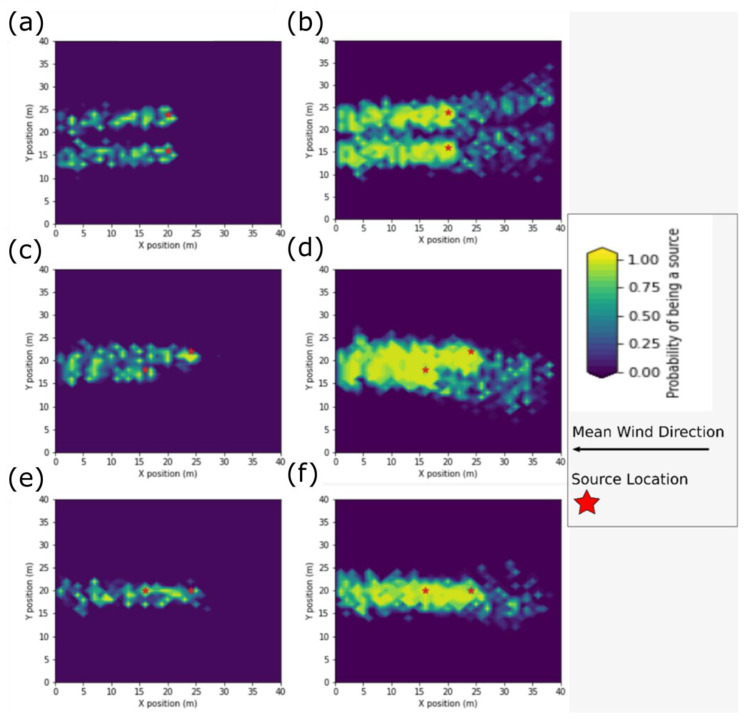
Occupancy grid map at the normal parameter level for (**a**) the IP algorithm with the crosswind source configuration, (**b**) the DS algorithm with the crosswind source configuration, (**c**) the IP algorithm staggered source configuration, (**d**) the DS algorithm staggered source configuration, (**e**) the IP algorithm with the inline source configuration, and (**f**) the DS algorithm with the inline source configuration.

**Figure 6 sensors-23-04799-f006:**
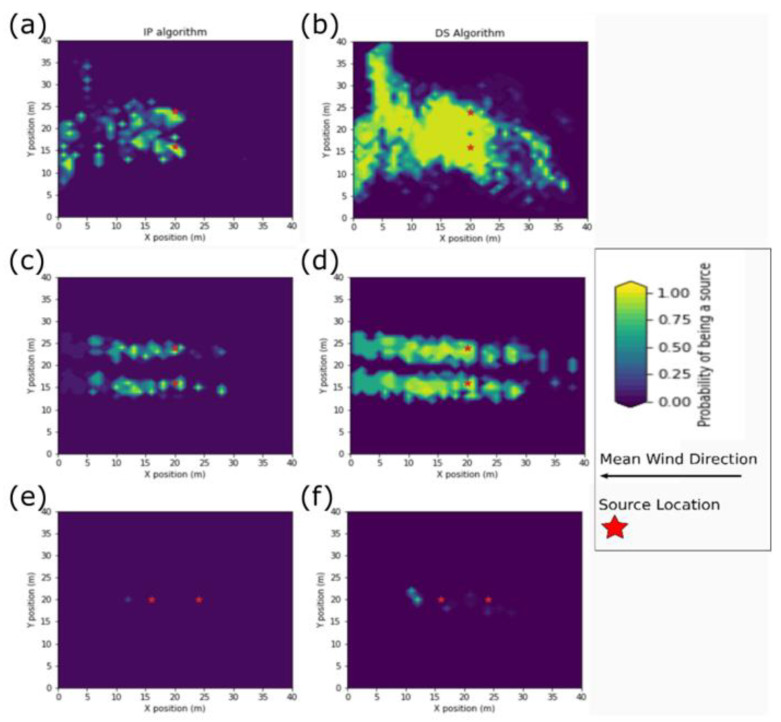
The occupancy grid map for (**a**) the IP algorithm with the crosswind source configuration with the low wind speed parameter level, (**b**) the DS algorithm with the crosswind source configuration with the low wind speed parameter level, (**c**) the IP algorithm with the crosswind source configuration with the high wind speed parameter level, (**d**) the DS algorithm with the crosswind source configuration with the high wind speed parameter level, (**e**) the IP algorithm with the inline source configuration with the low release rate parameter level, and (**f**) the DS algorithm with the inline source configuration with the low release rate parameter level.

**Figure 7 sensors-23-04799-f007:**
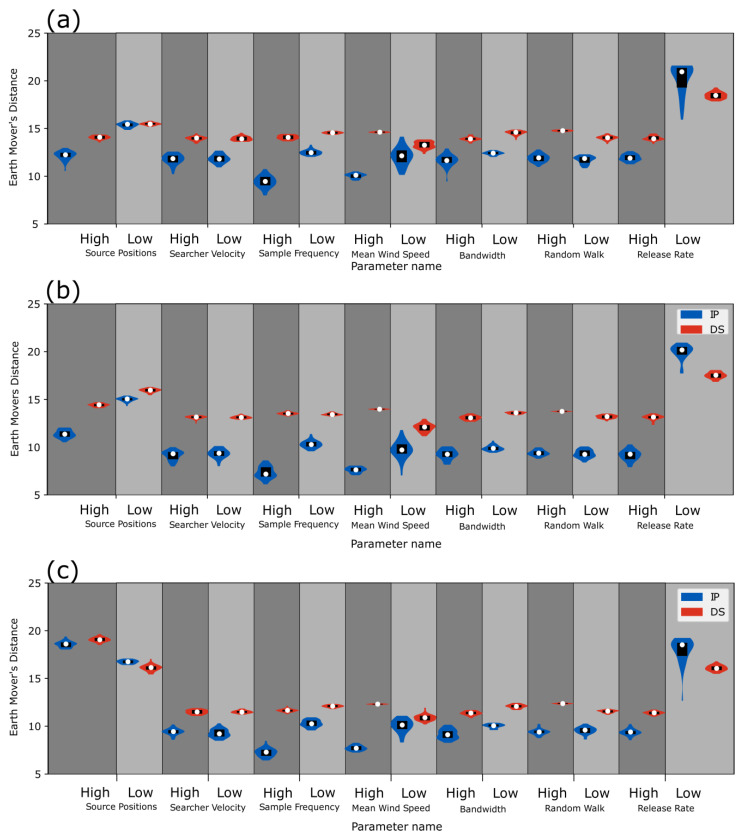
(**a**) Violin plots of EMD scores for the staggered source configuration, (**b**) violin plots of EMD scores for the inline source configuration, and (**c**) violin plots of EMD scores for the crosswind source configuration.

**Table 1 sensors-23-04799-t001:** List of parameters tested and their values at their low, normal and high levels.

Parameter	Low	Normal	High
Source positions	downwind	middle	upwind
Searcher velocity	0.5	1	2
Sample frequency	0.2	1	2
Mean wind speed	0.25	1	5
Bandwidth	0.01	0.15	0.5
Random walk	0.01	0.1	0.5
Release rate	0.01	20	80

**Table 2 sensors-23-04799-t002:** Comparison of the EMD between IP and DS algorithms for each parameter using the Wilcoxon signed-ranked test.

Wilcoxon Signed-Ranked Test									
Source Config	Parameter Level		Source Position	Searcher Velocity	Sample Frequency	Wind Speed	Bandwidth	Random Walk	Release Rate
Crosswind	Low	DS–IP	−0.61	2.28	4.36	0.77	2.03	1.99	−2.47
		*p*-value	2.35 × 10^−6^	1.73 × 10^−6^	1.73 × 10^−6^	1.49 × 10^−5^	1.73 × 10^−6^	1.73 × 10^−6^	3.72 × 10^−5^
	High	DS–IP	0.44	2.06	1.85	4.60	2.28	2.98	2.02
		*p*-value	1.92 × 10^−6^	1.73 × 10^−6^	1.73 × 10^−6^	1.73 × 10^−6^	1.73 × 10^−6^	1.73 × 10^−6^	1.73 × 10^−6^
Staggered	Low	Low–High	0.02	2.10	4.61	1.12	2.19	2.18	−2.48
		*p*-value	6.56 × 10^−2^	1.73 × 10^−6^	1.73 × 10^−6^	1.49 × 10^−5^	1.73 × 10^−6^	1.73 × 10^−6^	2.60 × 10^−5^
	High	DS–IP	1.84	2.16	2.08	4.52	2.26	2.85	2.04
		*p*-value	1.73 × 10^−6^	1.73 × 10^−6^	1.73 × 10^−6^	1.73 × 10^−6^	1.73 × 10^−6^	1.73 × 10^−6^	1.73 × 10^−6^
Inline	Low	DS–IP	0.92	3.76	6.35	2.40	3.72	3.96	−2.63
		*p*-value	1.73 × 10^−6^	1.73 × 10^−6^	1.73 × 10^−6^	1.73 × 10^−6^	1.73 × 10^−6^	1.73 × 10^−6^	1.92 × 10^−6^
	High	DS–IP	3.08	3.85	3.14	6.36	3.82	4.38	3.89
		*p*-value	1.73 × 10^−6^	1.73 × 10^−6^	1.73 × 10^−6^	1.73 × 10^−6^	1.73 × 10^−6^	1.73 × 10^−6^	1.73 × 10^−6^

**Table 3 sensors-23-04799-t003:** Comparison of the EMD between the low and high parameter level for each parameter using the Mann–Whitney U test.

Mann–Whitney U Test									
Source Config	Parameter Level		Source Position	Searcher Velocity	Sample Frequency	Wind Speed	Bandwidth	Random Walk	Release Rate
Crosswind	IP	Low–High	−1.86	−0.23	2.95	2.41	0.95	0.21	9.14
		*p*-value	3.02 × 10^−11^	1.54 × 10^−1^	3.02 × 10^−11^	3.02 × 10^−11^	5.46 × 10^−9^	5.55 × 10^−2^	3.02 × 10^−11^
	DS	Low–High	−2.91	−0.02	0.44	−1.42	0.70	−0.79	4.66
		*p*-value	3.02 × 10^−11^	8.77 × 10^−1^	3.82 × 10^−10^	3.02 × 10^−11^	3.34 × 10^−11^	3.02 × 10^−11^	3.02 × 10^−11^
Staggered	IP	Low–High	3.21	−0.04	3.01	2.04	0.77	−0.07	9.06
		*p*-value	3.02 × 10^−11^	8.07 × 10^−1^	3.02 × 10^−11^	1.33 × 10^−10^	7.69 × 10^−8^	1.37 × 10^−1^	3.02 × 10^−11^
	DS	Low–High	1.38	−0.09	0.48	−1.36	0.70	−0.74	4.54
		*p*-value	3.02 × 10^−11^	7.73 × 10^−1^	8.99 × 10^−11^	3.02 × 10^−11^	1.29 × 10^−9^	3.02 × 10^−11^	3.02 × 10^−11^
Inline	IP	Low–High	3.69	0.05	3.10	2.09	0.62	−0.12	10.91
		*p*-value	3.02 × 10^−11^	1.76 × 10^−1^	3.02 × 10^−11^	8.10 × 10^−10^	2.49 × 10^−6^	4.20 × 10^−1^	3.02 × 10^−11^
	DS	Low–High	1.54	−0.03	−0.11	−1.87	0.53	−0.55	4.39
		*p*-value	3.02 × 10^−11^	6.73 × 10^−1^	1.99 × 10^−2^	3.02 × 10^−11^	3.16 × 10^−10^	3.02 × 10^−11^	3.02 × 10^−11^

## Data Availability

The data presented in this study are available on request from the corresponding author.

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
