# Peer review of "A Comparison of Multiple Odor Source Localization Algorithms"

_sensors, 2023, doi:10.3390/s23104799_

Round 1

Reviewer 1 Report

The authors comparied the source location algorithms with mobile sensor based on simulations data. It is interesting. However, there are still some issues which should be improved before publication, as shown with following:

1. The literature review is not enough. The authors just discussed some reearches before 5 years ago. Some new researches have not been mentioned. E.g. the authors said there were just two algorithms for MOSL, which is not correct. So, more recent researches should be added in the manuscript.

2. The information about the simulation scenarios and simulation data  is absent. The authors should added more explanation about it.

3.  The experiment test is necessary to verify the performance of the location algorithm besides the simulation scenarios.

Reviewer 2 Report

Reviewer’s comments:

1.     In the manuscript, the cited references should be exhibited in sequence in “Introduction” section.

2.     The exploration of literatures about localization tasks of chemical source should be increased and enhanced in “Introduction” section.

3.     On page 2, it is suggested that the last paragraph can be considered to delete for the writing style of journal.

4.     On page 6, the originating of Table 1 should be clearly described why it could be as the environmental parameters.

5.     In section 2, the description of IP algorithm by using the probability theory as its belief representation for each cell in the occupancy map should cite the related literatures for the introducing the performance compared with DS algorithm.

6.     In Fig.3, the variables should be indicated and defined in the content for describing the configuration of the MOSL simulation in detail.

7.      On the line 337 of page 10, the equation (5) should be sequentially marked in equation “(7)”.

8.       In “Discussion” section, the difference of the simulated results between Independent Posteriors (IP) and Dempster-Shafer (DS) algorithms should be compared with the identified actual sources, the identifying source locations, the performance and limitations, and so on by using an obvious Table.

9.     The “5.1 future work” section is suggested to delete in the manuscript, which is not relationship with this research objective and research results in this journal paper.
